

# LARGE 0.2.0: 2D numerical modelling of geodynamic problems

Nicola Creati[1,*] and Roberto Vidmar[1]

[1]National Institute of Oceanography and Applied Geophysics - OGS, Borgo Grotta Gigante 42/c, Sgonico, Trieste, Italy

**Correspondence:** Nicola Creati (ncreati@inogs.it)

**Abstract.** We present here *LARGE* 0.2.0 (Lithosphere AsthenospheRe Geodynamic Evolution) a geodynamic modelling Python package that implements a flexible and user friendly tool for the geodynamic/modelling community. It simulates 2D large scale geodynamic processes by solving the conservation equations of mass, momentum, and energy by a finite difference method with the moving tracers technique. *LARGE* uses advanced modern numerical libraries and algorithms but unlike common simulation code written in Fortran or C this code is written entirely in Python. Simulations are driven by configuration files that define thoroughly the lithologies and the parameters that distinguish the model. Documentation for them and for all the modules is included in the package together with a complete set of examples and utilities. The package can be used to reproduce results of published studies and models or to experiment new simulations. *LARGE* can run in serial mode on desktop computers but can take advantage of MPI to run in parallel on multi node HPC systems.

## 1 Introduction

Numerical geodynamics aims to propose and to verify hypothesis and transform observations to prediction. Numerical simulations help to constrain the spatial and temporal evolution of the surface and the internal structure of the Earth since some quantities cannot be measured, In the late 1960s the theory of plate tectonic pushed geodynamics to move from a descriptive to a quantitative view (McKenzie and Parker, 1967; Minear and Toksöz, 1970). Luckily advancements in geomechanics, numerical methods, computational resources, and the increase of observations and measurements on the surface and the deep Earth helped to create increasingly sophisticated simulations (Gerya, 2019). Complex rheologies (Gerya, 2019), cylindrical and spherical geometries (Stadler et al., 2010), mineral physics (Tackley, 2000) and magma transport (Katz and Weatherley, 2012) can be considered in modern geodynamic modeling. Geodynamic simulation can be accomplished using commercial software (Pascal and Cloetingh, 2002; Jarosinski et al., 2011), but there are also several other tools available in the literature, both open and closed source, that do the same job and add new numerical methods and algorithms. Some are well documented (Moresi et al., 2007; Wilson et al., 2017) but the user cannot change anything unless he understands the programming language used for coding and all the math involved. All these tools are written either in Fortran, C or C++ and integrate some well established libraries that provide top-notch numerical methods for the solution of differential equations. Some software packages expose a scripting interface (Moresi et al., 2007; Aagaard et al., 2013; Wilson et al., 2017) to satisfy user experience and push users to investigate their own problems.





*LARGE* 0.2.0 (Lithosphere AsthenospheRe Geodynamic Evolution) is a numerical geodynamic simulation software entirely written in Python (van Rossum, 1995), and released under the MIT license. Python is an object oriented interpreted language and one of the preferred programming/scripting languages by data scientists for prototyping, visualization, and running data analyses on small and medium sized data sets. The main reasons of its success are that it's simple, free, friendly with other

languages, object oriented and has a lot of available libraries (packages in Python jargon) that extend Python domain of enforcement. From its introduction in the '90 several packages has been developed to improve its numerical computation capabilities (Walt et al., 2011; Virtanen et al., 2020). It is one of the most popular programming languages (3th according to the TIOBE Index, https://www.tiobe.com/tiobe-index/) and hundreds of tutorials can be easily found on the web. The language is self-sufficient, comes out-of-the-box ready to use, with everything that is needed. Being interpreted it's inherently slow if

compared with other compiled languages but this limit can be overcome with good programming skills. Being object-oriented and self documented, Python code can be easily reused and tailored to different applications.

*LARGE* is driven by a hierarchical structure that defines the geometry, the spatial distribution of the lithologies with their characteristics, the initial temperature and other details. The model is defined by a single text configuration file that controls its evolution in time. *LARGE* has been developed on Linux desktop computers with the target of fast HPC systems. It can

run in serial mode for small models on limited power systems and in parallel mode taking full advantage of multi-core / multi-node architecture to solve large, complex and detailed two-dimensional problems. Parallel execution is mandatory since the requested resolution of numerical models increased in the last twenty years but the maximum speed of a single CPU is by now at its physical limit (just above 5 GHz) due to cooling issues. The solution found to overcome this limit was the development of multi-core CPU, multi-node computers and parallel / concurrent algorithms. Researchers can now run their

complex simulations on clusters of computing nodes of different architectures (shared memory systems, distributed memory systems and distributed shared memory systems) but to take advantage of this architecture parallel algorithms and software packages must be written. Parallel algorithms break the computational issues into several discrete parts that have to be computed separately and all these parts should operate concurrently. In order to solve the initial problem therefore a way to coordinate every operation and to communicate intermediate data is needed, some algorithms must be rewritten and some other must be

created. Python, in a parallel environment, preserves all its advantages reducing the effort to create and to debug an efficient parallel code. The next sections will describe the numerical background of *LARGE*, the Python packages used, how the code has been tailored, the tools developed to help users in creating and running their simulations, and some examples. At the end we will address possible extensions and work in progress.

## 2  Numerical methodologies

*LARGE* implements a two dimensional visco-elasto-plastic simulation code that is based on the method proposed by Gerya and Yuen (2003, 2007) and then extended and checked by Deubelbeiss and Kaus (2008); Duretz et al. (2011). It implements a conservative finite difference method coupled with a tracers (sometimes called also particles or markers) in cells (TIC) algorithm (Harlow and Welch, 1965). *LARGE* is not a generic differential equation solver but solves the conservation of mass





(eq. 1), momentum (eq. 2) and energy (eq. 3) in a 2-D continuum assuming material incompressibility:

$$\frac{\partial v_i}{\partial x_i} = 0 \tag{1}$$

$$\frac{\partial \sigma_{ij}^{'(t)}}{\partial x_j} - \frac{\partial P}{\partial x_i} + \rho g_i = 0 \tag{2}$$

$$\rho C_p \frac{DT}{Dt} = -\frac{\partial}{\partial x_i}\left(k\frac{\partial T}{\partial x_i}\right) + H_s \tag{3}$$

where the quantities are denoted in Table 1. $DT/Dt$ is the material time derivative of $T$ and the repeated indices $i,j$ mean summation (Einstein notation). Eqs. (1, 2) are discretized on an Eulerian grid with a conservative finite difference scheme on a fully staggered mesh, that can be either regular or irregular (Duretz et al., 2011; Gerya and Yuen, 2007). Eq.(3) is solved in a Lagrangian frame using a Forward-Time Central-Space (FCTS) scheme. The stability of the solution, low diffusion, is achieved by testing the maximum temperature change between the current solution and the solution of the previous time step. If the change is greater than an imposed threshold the time step is split in several smaller time ranges and the equation repeatedly solved.

The TIC method handles advection and the movement of material properties in the velocity field calculated from the Stokes equations 1, 2. The advection uses a CorrMinMod algorithm (Jenny et al., 2001; Pusok et al., 2017) coupled with a time and spatial explicit fourth-order Runge-Kutta scheme.

The deviatoric stress components at the current time step, $\sigma_{ij}^{'(t)}$ in Eq.(2) is formulated from the rheological constitutive relationships. The rheological model uses a Maxwell linear stress-strain relationship:

$$\dot{\varepsilon_{ij}} = \dot{\varepsilon_{ij}}_{(viscous)} + \dot{\varepsilon_{ij}}_{(elastic)} \tag{4}$$

where:

$$\dot{\varepsilon_{ij}}_{(viscous)} = \frac{1}{2\eta}\sigma'_{ij} \tag{5}$$

$$\dot{\varepsilon_{ij}}_{(elastic)} = \frac{1}{2G}\frac{D\sigma'_{ij}}{Dt} \tag{6}$$

Following Moresi et al. (2003), the material derivative of eq. 6 can be discretized with backward finite difference in time to:

$$\dot{\varepsilon_{ij}}_{(elastic)} = \frac{1}{2G}\frac{\sigma_{ij}^{'(t)} - \sigma_{ij}^{'(t-\Delta t)}}{\Delta t} \tag{7}$$

Substituting eqs. 5, 7 in eq. 4 the stress strain-rate relationship becomes:

$$\sigma_{ij}^{'t} = 2\eta_{vp}\dot{\varepsilon_{ij}}Z + \sigma_{ij}^{t-\Delta t}(1-Z) \tag{8}$$





where $Z$, the visco-elastic factor, is defined as:

$$Z = \frac{G\Delta t}{G\Delta t + \eta_{vp}} \tag{9}$$

Hence, the deviatoric stress in eq. 2 depends on the effective viscosity, the strain-rate, the advection time interval and the deviatoric stress of the previous time step corrected for rotation and material diffusion. In this formulation the effective viscosity is modified by elastic stress and modulated by $Z$. The effective viscosity is modified by the rheological behaviour of rocks such as non-Newtonian material creep and plasticity:

$$\begin{cases} \eta_{vp} = \eta & \text{when } \sigma_{II} < \sigma_{yield} \\ \eta_{vp} = \eta \frac{\sigma_{II}}{2\eta\varepsilon_{II(plastic)} + \sigma_{II}} & \text{for } \sigma_{II} = \sigma_{yield} \end{cases} \tag{10}$$

The effective viscosity of rocks depends on rheological parameters measured in laboratory experiments, temperature, pressure and stress according to the following relationship (Ranalli, 1995):

$$\eta = \frac{F}{A_D} \sigma_{II}^{(1-n)} e^{\frac{E_a + V_a P}{RT}} \tag{11}$$

where $F$ is a dimensionless correction term that depends on the type of experiment used to measure the rock rheological parameters and lets to transform the equation in tensorial format (Ranalli, 1995; Gerya and Yuen, 2007). Furthermore the effective viscosity can be reduced by plastic deformation according to the Drucker-Prager failure criterion:

$$\sigma_{yield} = C + P \sin(\varphi) \tag{12}$$

Plasticity is crucial in the deformation of the lithosphere since it contributes to shear localization and the enucleation of several patterns of shear zones. The Stokes equations is non linear and the solution may not converge (Spiegelman et al., 2016) since in eq. 2 the deviatoric stress depends on viscosity and viscosity depends on the rheological equations and their parameters. *LARGE* implements an iterative algorithm (Gerya, 2018, 2019) to solve this issue.

A chain of tracers (Van Keken et al., 1997) represents and mimics the evolution of the surface topography. The chain is advected at each step of evolution according to the velocity field calculated by the Stokes equations. Additionally the surface can be deformed by smoothing processes modelled by a standard diffusion equation (Pelletier, 2008):

$$\frac{\partial h}{\partial t} = \kappa \frac{\partial^2 h}{\partial x^2} \tag{13}$$

The equation is solved using the Cranck-Nicolson scheme to avoid numerical diffusion. This method is unconditionally stable therefore no time step iteration is needed.

In the TIC method quantities carried by the tracers are interpolated from the tracers cloud to the staggered grid of nodes through bilinear interpolation according to a 4-Cell or 1-Cell scheme (Duretz et al., 2011).

## 3   Code design

*LARGE* was written from scratch, with an object-oriented structure, to be readable and reusable. The computation complexity of numerical simulation has been abstracted in classes that can be subclassed to add more features. *LARGE* can run in two



**Table 1.** Abbreviations and units of the quantities used in *LARGE*

| Symbol | Definition | Units |
|---|---|---|
| $v_i$ | Velocity field with components $(v_x, v_y)$ | $\mathrm{m\,s^{-1}}$ |
| $\sigma_{ij}'^t$ | Deviatoric stress tensor components | Pa |
| $\sigma_{yield}$ | Plastic yield stress | Pa |
| $P$ | Pressure | Pa |
| $\rho$ | Density | $\mathrm{kg\,m^{-3}}$ |
| $g_i$ | Gravity acceleration components $g_x, g_y$ | $\mathrm{m\,s^{-2}}$ |
| $C_p$ | Isobaric heat capacity | $\mathrm{J\,kg^1\,K^1}$ |
| $T$ | Temperature | K |
| $k$ | Thermal conductivity | $\mathrm{W\,m^{-1}\,K^{-1}}$ |
| $H_s$ | Heat sources | $\mathrm{W\,m^{-3}}$ |
| $\eta_{vp}$ | Visco-plastic viscosity | Pa s |
| $\eta$ | Effective viscosity | Pa s |
| $\dot{\varepsilon_{ij}}$ | Deviatoric strain rate components | $\mathrm{s^{-1}}$ |
| $Z$ | Visco-elastic factor | |
| $G$ | Shear/Rigidity modulus | Pa |
| $A_D$ | Power law constant | $\mathrm{MPa^{-n}\,s^{-1}}$ |
| $E_a$ | Power law activation enthalpy | $\mathrm{kJ\,mol^{-1}}$ |
| $V_a$ | Power law activation volume | $\mathrm{m^3}$ |
| $n$ | Power law exponent | |
| $R$ | Gas constant | $\mathrm{J\,mol^{-1}\,K^{-1}}$ |
| $\sigma_{II}$ | Second invariant of stress | Pa |
| $C$ | Rock cohesion | MPa |
| $\varphi$ | Internal friction angle | rad |
| $h$ | Topography | m |
| $\kappa$ | Surface diffusivity | $\mathrm{m^2\,s^{-1}}$ |

modes: serial for small models that fit on a single computer and parallel, through an executable called *LARGEP*, for large models that cannot fit in a single computing node memory or to speed up the execution taking advantage of the multi core architecture of modern CPU or using HPC systems.

## 3.1 Dependencies

*LARGE* makes use of the following packages freely available from the Python ecosystem:

– **Numpy** (Walt et al., 2011) implements an array data structure and fast vectorized basic arithmetic computations. It's a standard in the scientific Python community and it is integrated in several scientific Python packages. Algorithms that





process a single value at the time can be converted to operate on a set of values at the same time. Vectorization implies the substitution of for-loop construct with vectorized methods provides by Numpy. Sometimes vectorized code looses its readability but the speed improvement can be awesome.

– **Scipy** (Virtanen et al., 2020) is based on Numpy and provides fast routines for numerical integration, sparse matrix representation, interpolation, statistics, optimization, linear algebra, statistics and more.

  – **QCOBJ** (Vidmar and Creati, 2018) was created during the development of *LARGE* to allow the definition of physical quantities with units of measurement in configuration files freeing the users from possible unit conversions errors.

  – **H5py** (Collette, 2013) is a bind to the High-performance data management and storage suite (The HDF Group, 1997). It
provides a set of I/O operation on a file format designed to store and organize large volumes of data.

  – **Numba** (Lam et al., 2015) is a just-in-time compiler for Python that speeds up array operations. The most common way to use Numba is through its collection of decorators that can be applied to functions to instruct Numba to compile them. When a call is made to a Numba decorated function it is compiled to machine code "just-in-time" for execution and the code can subsequently run at native machine code speed.

– **colorAlphabet** (Vidmar, R., 2018) adds color to text written on a terminal using Paul Green-Armytage color alphabet (Green-Armytage, 2010). *LARGE* uses it to differentiate logging messages originated from different processes and their severity.

  – **PySide2** (The Qt Company) (or PyQt5 (Riverbank Computing Limited)) is not strictly required to run *LARGE* but is necessary to take advantage of the graphic utilities included in the package.

For *LARGEP* also the following packages are needed:

  – **MPI4py** (Dalcín et al., 2005) provides Python bindings for the Message Passing Interface (MPI) standard. It allows a Python program to execute on a computer cluster of different computing nodes using MPI to communicate.

  – **Petsc4py** (Dalcin et al., 2011) allows the Portable Extensible Toolkit for Scientific Computation (PETSc) to be used in Python. PETSc is a suite of data structure and routines able to solve discrete algebraic equations, usually arising from
the discretization of partial differential equations. It has many features, including scalable, parallel linear, and non linear solvers, which are used and tested in numerous other models and scientific applications.

  – **miniga4py** (Vidmar, R., 2020b) is a subset of the binding to the Global Array Toolkit (GA) (Nieplocha et al., 2006) tailored to operate in *LARGE*. GA implements the partitioned global address space (PGAS) parallel programming model and allows the creation and management of arrays defined in the context of a distributed data structures. GA arrays
are global and can be used as if they were stored in a shared memory environment. All details of the data distribution, addressing, and data access are encapsulated in the global array objects. The basic shared memory operations supported



include get, put, scatter, and gather. These operations are truly one-sided/unilateral and will complete regardless of any action taken by the remote process(es) which own(s) the referenced data.

– **h5pyp** (Vidmar, R., 2020a) provides Python bindings for the parallel version of the High-performance data management
and storage suite (The HDF Group, 1997).

## 3.2 The configuration file

*LARGE* numerical simulations are driven by a single configuration file. This text file has a hierarchical structure with many parameters related to the model and its evolution in time and five sections that define other features in detail.

– The **Mesh** section defines the geometry of the domain, the size and the layout of the grid.

– The **Lithologies** section defines the different type of rocks that make the model, their spatial arrangement and their rheological behaviour through twenty parameters for each of them. Each lithology is bounded by polygons that are defined in this section.

– The **Geothermic Model** section defines areas of the model, delimited by polygons, with different initial temperature. *LARGE* provides some of the most common geothermal models: conductive, radiogenic, half-space, plate. These are
used in the initialization of the model to calculate the initial temperature of the Lagrangian tracers. As the definition of the initial temperature distribution can be complex, *LARGE* allows the user to define his own model through a Python module provided at run time.

– The **Boundary Conditions** section defines the behaviour of the Stokes and Heat equations at the domain boundaries. By now *LARGE* does not support every possible boundary condition but only few of the most commonly used in the
geodynamic literature. Stokes equations can manage free-slip and no-slip conditions. Moreover a prescribed velocity can be set on free-slip boundaries and the edges can also move (moving wall condition). Heat equation supports Dirichlet and Neumann conditions.

– The **topography** section defines the parameters for the creation and evolution of a 1D chain of tracers with a fixed elevation. An external file can be specified to load a height profile if the topography is not constant.

Every physical quantity related to the physics of the simulation must be expressed with its unit of measurement. *LARGE*, while parsing the configuration file, checks its dimensionality and converts it into the right units required by the code.

## 3.3 Program execution

*LARGE* starts its execution creating a Model object instance, parsing the configuration file, and initializing some other objects that will be used during the evolution (Fig. 1):

– The **Mesh** defines the model domain geometry, dimensions, nodes density and evaluates domain deformations during the simulation.



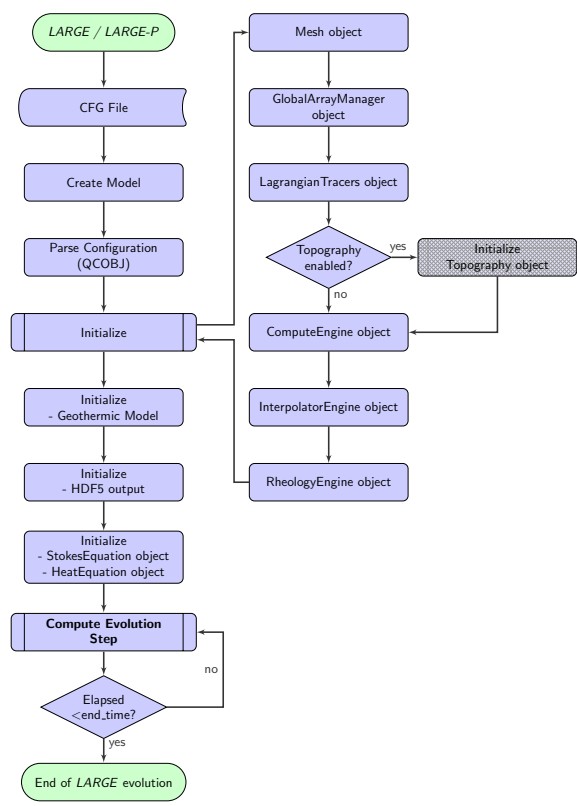

**Figure 1.** Flowchart of *LARGE*.

- The **GlobalArrayManager (GAM)** is trivially an array container when running in serial mode but allocates parallel dense bidimensional arrays among all available processors when running *LARGEP*. It provides a high level interface to some arithmetic operations on parallel arrays and methods to modify only parts of the allocated arrays. Based on
miniga4py, GAM hides all the complexity of managing the distributed arrays between the computing nodes making them available as Numpy arrays. All details of the data distribution, addressing, and data access are encapsulated in global arrays objects.

- The **LagrangianTracers** object manages the tracers used to record quantity changes in the moving medium. Tracers are a sparse cloud of points, built with a Halton distribution (Halton, 1964), that in parallel mode is distributed among all the
processors. This object takes care of all communication between the processors regarding tracers distribution, advection, deletion, repopulation and ghosts handling through algorithms that use Numpy and MPI.

- The **Topography** object, whose usage is activated in the configuration file, provides methods to evaluate how the top of a model deforms. This profile is represented as a 1D chain of tracers advected by the velocity calculated solving the Stokes equations. It is furthermore deformed by an erosional process described by a diffusion equation.





**Figure 2.** Flowchart of the evolve and VPI algorithms used in *LARGE*.

– The **ComputeEngine** object groups operations on global arrays or tracers that do not involve any interpolation of quantities from tracers to node or node to tracers.

    – The **InterpolatorEngine** object instead implements all algorithms for the interpolation of quantities from tracers to node and back.

    – The **RheologyEngine** object uses rocks rheological parameters defined in the configuration file to calculate the viscosity

of rocks according to a power-law model coupled with the Drucker-Prager plastic failure criteria.

Following the schema in Fig. 1 the **Geothermic Model** is initialized, i.e. the initial temperature of every tracer is computed according to the model defined in the configuration file (or supplied by the user) and an **HDF5** output file is created. This file





contains the whole configuration used in the current simulation, information about the software environment and about the model evolution. The **StokesEquation** object creates and stores the coefficients needed to solve the Stokes equations with the

boundary conditions defined in the configuration file. A fast vectorial algorithm without for-loops constructs fills the systems. *LARGE* serial uses Scipy sparse solver while *LARGEP* adopts PETSc to distribute the system of equations among all available processors. *LARGEP* implements a direct parallel solver to solve the system of equations and by default it uses the MUMPS (Amestoy et al., 2001, 2006) library but the type of solver and the parameters used for the solution can be set in the configuration file. The **HeatEquation** object shares the same structure with the StokesEquation object but solves the energy equation.

After the initialization phase is completed, the model evolution can start, computing every time step, until the *time_end* or *step_end* defined in the configuration is reached. Users that want to implement their own *evolve* function can supply it as a module at run time. The computing of a single evolution step is described in Fig. 2. During model evolution all the tracers move according to the physical laws for the time defined by the *timestep* time interval parameter.

Running *LARGEP* the cloud of Lagrangian tracers is distributed among all processors divided in rows and columns so that

every processor holds only a part of them but an extra number of tracers from the neighboring processors is needed to compute values at the borders. These are called *ghost* tracers. Ghost tracers are added at every loop iteration since tracers move during the evolution and might migrate from one processor to another.

The visco-plasticity iteration loop (VPI) starts with the interpolation of scalar quantities defined on the tracers and continue solving the Stokes equations. The Stokes equations system can be ill-posed since the problem is strongly non-linear and could

not converge masking the yielding condition (Spiegelman et al., 2016; Gerya, 2018). This issue can be overcome iterating on the domain grid nodes by reducing the evolution time interval if the yielding error is greater than a user defined threshold or if too many iteration occur without any significant reduction of it. The Courant time interval criterium has been added to the VPI loop to reduce the time interval if one the maximum values of the velocity components ($v_x$ and $v_y$) from the solution of the Stokes equations is too high. If for any reason the time interval has been reduced, VPI must restart with the new time interval

value but if too many iteration occurred, VPI must restart anyway reducing it. VPI must be done since stresses, strain rates and viscosity depend on the velocity components computed solving the Stokes equations. At every VPI step viscous viscosity is calculated from the creep rheological model, velocity and pressure are calculated solving the Stokes equations, and plastic yielding, visco-plastic viscosity, stresses and strain rates are calculated too.

Once the visco-plasticity has been calculated, the time step evolution continues with the solution of the Heat equation, the

advection of tracers with the CorrMinMod algorithm (Jenny et al., 2001; Pusok et al., 2017) and the rotation of stresses since the materials are elastic. Topography, if defined, is updated and all the physical quantities are interpolated back to the tracers. The boundary conditions are updated if the domain is deformable (i.e. boundary walls can move), the mesh geometry and the grid structure are updated according to the computed velocities. Next, if running *LARGEP*, the tracers must be redistributed among the processors. The *tracersReseedGlobal* parameter of the configuration file controls the reseeding of the tracers cloud.

If the total number of tracers in the domain reduces below this threshold the whole cloud of points is replaced by a new one. The *tracersReseedLocal* parameter controls instead the reseeding if the number of tracers in a single cell reduces below threshold.





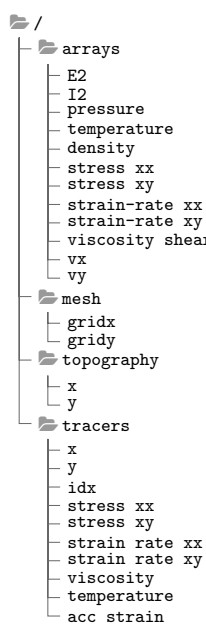

**Figure 3.** *Large* HDF5 step file structure. The arrays group node is filled by 2D global arrays. The tracers group contains 1D tracers quantities arrays. Mesh and Topography contain 1D arrays. The structure of this file is modular and other quantities, 2D or 1D, can be added to their respective group by adding their name in the configuration file.

At this point all the quantities have been calculated and can now be saved to an HDF5 file. This *step* file is a snapshot of the model evolution at a certain time from the beginning and is the output of the simulation. Step files have a hierarchical structure (Fig. 3) with the following HDF5 groups: arrays of the physical quantities defined on the Lagrangian grid (2D arrays), tracers (1D arrays), mesh and optionally topography. Each group contains the arrays as datasets.

As the size of these files can be large and the I/O operations slow down the model evolution, the configuration file provides options to choose the quantities to save and how often to save them. The quantities saved by default are necessary to allow *LARGE* to continue a previously interrupted simulation, in append mode, re-starting from the last valid step.

*LARGE* uses an MPI compliant logging system that helped during the developing stage since it lets to follow the single phases of the evolution in every processor when running *LARGEP*.

## 4 Command line utilities

*LARGE* integrates some tools that can be executed from the command line.

**cfg_gui** is a graphical tool that helps the creation and tuning of the configuration file which is usually a long and error prone procedure. It shows the hierarchical structure of the configuration file with a vertical tree representation. Each field of the tree can be edited to change the value. Moreover it is possible to compare two and even more configuration files.





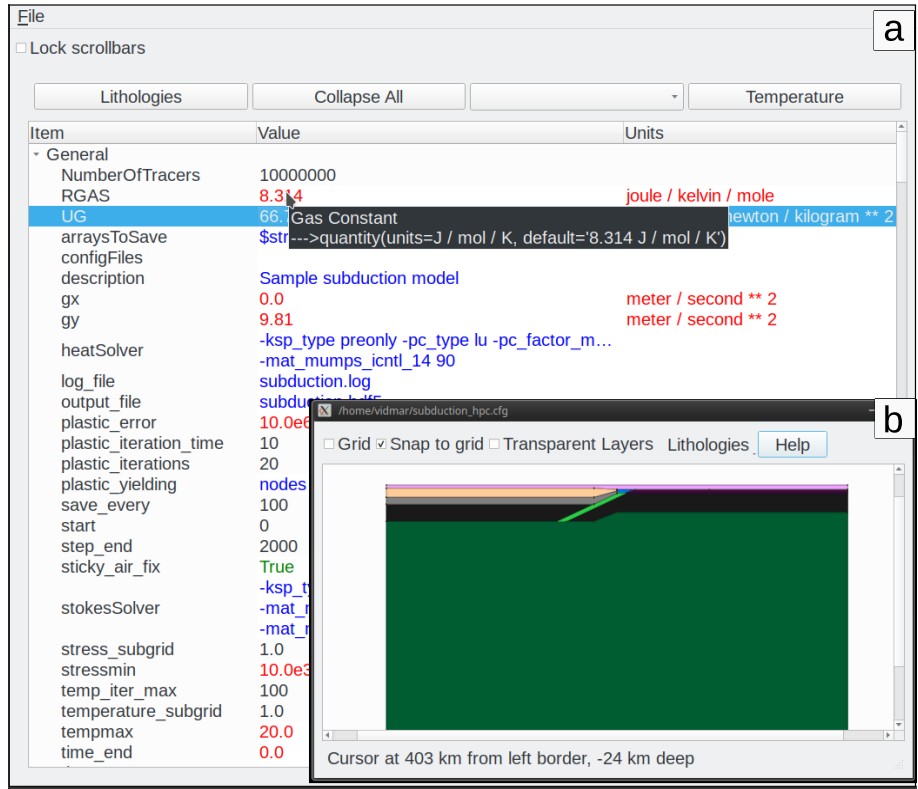

**Figure 4.** (a) Large_cfg_gui shows the hierarchical tree structure of a *LARGE* configuration file. Every parameter can be edited. Moreover the tool lets to compare up to three different configuration file and highlight the differences between them. (b) Geometric representation of the polygons defined in the Lithologies section of the configuration file. The shape of Each polygon can be interactively modified as in standard vector graphics programs.

This proved to be useful when only few parameters change between two simulations. It is based on the *cfggui* module provided by *QCOBJ* (Fig. 4a) and integrates a graphic editor designed to visualize and interactively edit the polygons defined in the configuration file (Fig. 4b).

**large2largep** This tool has been developed to automate the process of downloading, compiling and installing all the libraries
and Python packages required by *LARGEP*.

**largep** is the launcher of *LARGEP*. It accepts all the options of *LARGE* and also:

- sets the number of MPI process to use;

- kills hanging *LARGEP* process of simulations ended abnormally;

- executes every single *LARGE* processes in separate terminal windows, one for each process, to make debugging
easier.





**largeh** gives quick access to the HTML documentation included in the *LARGE* package. This command runs the default browser and allows quick retrieval of the information needed to prepare and run *LARGE* simulations. The configuration file reference, the package API reference, the description of the tools and the remaining documentation of the package, including the examples, are thus immediately available for consultation.

**largel** allows to visualize and search through *LARGE* log files that use ANSI escape codes to add color information to the text.

**largeinfo** is a GUI application created to display the content of the base output file of a *LARGE* simulation run . The command line used when the simulation was launched, the software versions of the most important packages used, the whole configuration file content, the last step done and the number of MPI processes used are some of the information provided.

## 5   Examples

The ability of *LARGE* to solve geodynamic problems can be tested with some examples inspired from well known models from the literature that are enclosed in the package. These examples are also the best way to practice with the configuration options.

### 5.1   Shear bands

Plastic failure is a common deformation mechanism in the upper lithosphere. The **shear bands** model consists of a visco-275 elasto-plastic material with a fixed viscosity of $10^{23}$ Pa · s and some viscous inclusions with a viscosity of $10^{17}$ Pa · s (Kaus, 2010) at random positions and depth (Fig. 5a). The configuration file for this model, *shear_bands.cfg* is located in the *cfg* folder of the *LARGE* source code repository. This simulation does not take in account the temperature therefore the heat calculation is switched off in the configuration file by setting the equation solver to *none*. For the same reason the geothermal model section is missing in the *cfg* file. The Stokes boundary conditions are set to *free slip* for all boundaries with prescribed inflow 280 velocities for the left and right boundaries and prescribed outflow velocities for the top and bottom boundaries. After 140 (Fig. 5b) evolution steps shear bands appear with an angle of around $26°$ that is close to the Coulomb angle of $45° \pm \theta/2$ where $\theta$ is the angle of friction. At step 180 (Fig. 5b) some shear bands die but if they reach a boundary they are reflected, as the boundary acts as a point to initiate a new shear band. This phenomenon happens because the VPI is enabled, if disabled, setting just one iteration for the VPI in the *cfg*, no shear bands will appear.

### 5.2   Plate Subduction

The setup of plate subduction models can be tricky since the lithological and the geothermal model may have many different subsections but even if the overall geometry can be complex, *large_cfg_gui* can help to draw the model. We will describe here two models:

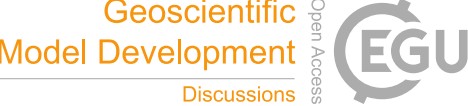



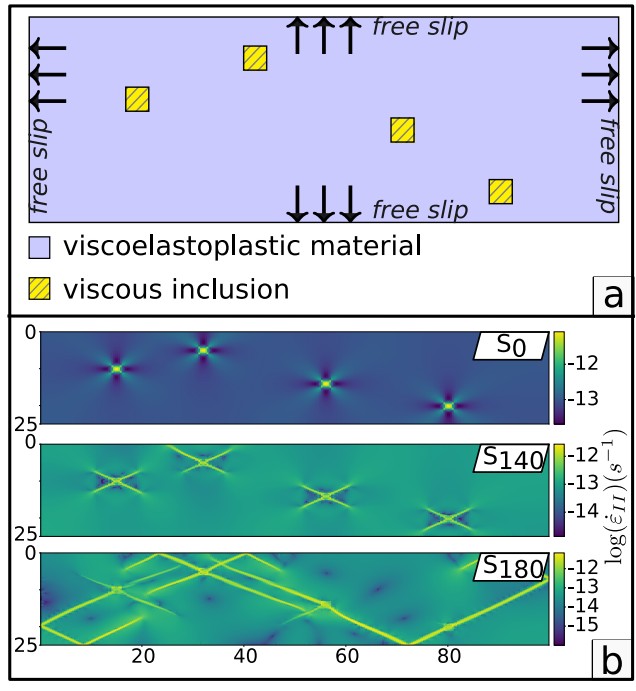

**Figure 5.** (a) Shear bands model setup; (b) Evolution at three time steps: 0, 140 and 180. At step 140 shear bands start to grow at an angle close to the Coulomb angle. At step 180 some shear bands reach the boundaries and are reflected. The enucleation of shear bands can happen only if the VPI is enabled in the configuration file otherwise no shear band will appear.

- the **plates converge** model which is the subduction of an oceanic plate under a continental one driven by the oceanic plate converge velocity (Gorczyk et al., 2007). The configuration file for this model, *subduction.cfg* is located in the *cfg* folder of the *LARGE* source code repository.

- the **slab retreat** model which is the subduction and retreat of an oceanic plate under a younger oceanic one driven by the buoyancy of the older. The configuration file for this model, *slab_retreat.cfg* is located in the *cfg* folder of the *LARGE* source code repository.

### 5.2.1 Plates converge

The model (Fig. 6a) consists of a continental plate with a 35 km crust, composed by 20 km of upper crust wet quarzite (Ranalli, 1995) and 15 km of lower crust anorthosite (Ranalli, 1995), with the base of the continental lithosphere fixed at 80 km. The lid and the asthenosphere adopt a dry olivine rheology (Ranalli, 1995). The oceanic plate has a lithosphere thickness of 60 km compatible with a thermal age of 30 My. The oceanic lithosphere contains a thin crust, of about 10 km, split in three layers of sediments, basalts and gabbroic rocks modelled by a wet quarzite and anorthosite rheology. The lithosphere-asthenosphere boundary (LAB) has a fixed temperature of 1330 °C. The Stokes boundary conditions are free slip for the left, right and



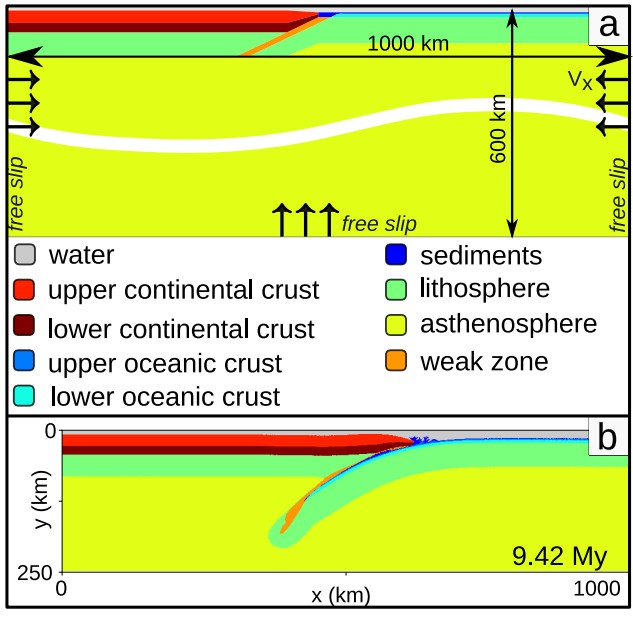

**Figure 6.** (a) Plate convergence subduction model setup. A velocity of -2 $\mathrm{cmy}^{-1}$ has been applied to the right border. Temperature has been fixed to 0 °C at the top and 1600 °C at the bottom boundary.; (b) Model evolution after 9.42 $\mathrm{My}$. The oceanic plate bends under the continental plate and the tip has penetrated down to 200 $\mathrm{km}$. The model shows a shortening of about 192 $\mathrm{km}$.

bottom edges while the top boundary free surface condition is approximated by a free slip one coupled by a thin, 7-10 $\mathrm{km}$, low viscosity layer of water. Numerical instabilities of the Stokes solution in the top layer is reduced by enabling the Stikcy-air approximation (Duretz et al., 2011) in the configuration file. Left, right and bottom edges can move and a shortening velocity of
-2.0 $\mathrm{cmy}^{-1}$ has been applied to the right edge. The heat equation is solved setting the Neumann condition at the left and right boundaries and the Dirichlet condition, with fixed temperatures of 0 °C at the top and 1600 °C at the bottom edges. The model extends for 1000 $\mathrm{km}$ in $x$, 600 $\mathrm{km}$ in depth ($y$ axis) and has 10 millions of Lagrangian tracers. A weak zone, approximated by a wet olivine rheology with no friction, is configured at the boundary between the oceanic and continental plate to promote the subduction start. The initial temperature distribution is different for the continental and oceanic zones. The initial continental
lithosphere temperature is described by a conductive model while the initial temperature of the oceanic lithosphere is calculated by a standard plate model (Parson and Sclater, 1977). VPI is enabled to ensure the occurrence of plastic shearing in the brittle parts of the lithosphere.

The weak zone triggers the subduction and after 9.4 $\mathrm{My}$ the tip of the bending slab reaches a depth of 200 $\mathrm{km}$ (Fig. 6b). The bending is favored by plastic deformation along shear bands in the upper portion of the subduction slab.



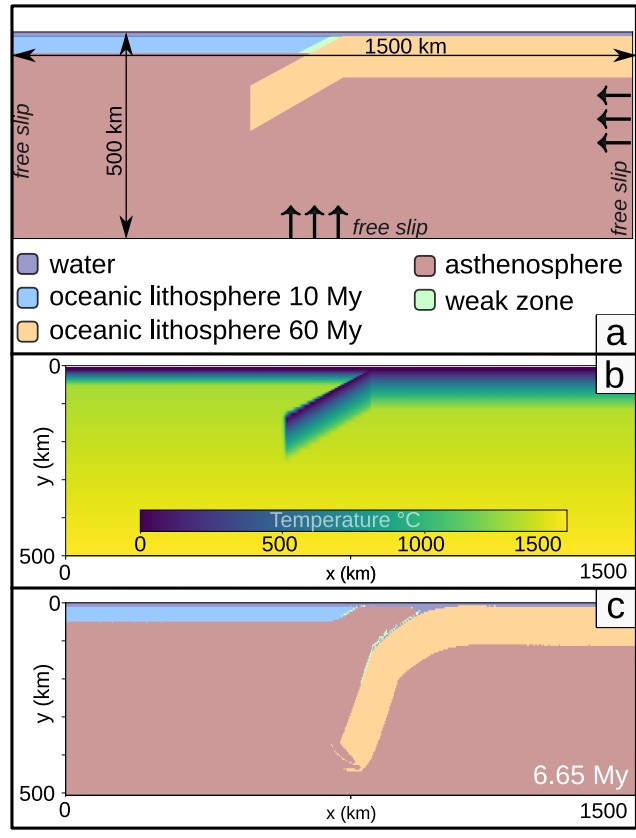

**Figure 7.** (a) Slab retreat model setup. No velocity has been applied to the edges of the model domain. Subduction is driven by the buoyancy of an older oceanic plate under a younger one. Temperature has been fixed to 0 °C at the top and 1500 °C at the bottom boundary. (b) Initial distribution of the temperature in the domain, calculated by an external user provided Python module, loaded at run time. (c) Model evolution after 6.65 My showing the progressive retreat of slab and upwelling of mantle material in the space widened between the two oceanic plates.

### 5.2.2 Slab retreat model

The slab retreat model (Fig. 7a) is an opportunity to show how to setup a user provided geothermal model module. Here, the subduction is driven by the buoyancy of the subducting plate. There are no prescribed velocities at the edges of the domain and the boundaries cannot move. The geometry is characterized by two oceanic plates of different ages, 10 My and 60 My, with the older one that is already subducting and has penetrated the asthenosphere. A weak zone, approximated by a wet olivine rheology with 0 friction, is configured at the boundary between the two oceanic plates to favour plates decoupling. In this model the subducting plate is penetrating the asthenosphere and temperature must be calculated in an inclined slab (Fig. 7b) but *LARGE* does not provide any function to calculate the initial temperature in complex geometries so a custom Python module was created and loaded at run time to compute the right temperature. The LAB temperature has been fixed at 1330 °C, and the heat equation is solved setting the Neumann condition at the left and right boundaries and the Dirichlet condition,





with fixed temperatures of 0 °C at the top and 1500 °C at the bottom edges. The domain extends for 1500 km in $x$, for 600 km in depth ($y$ axis) and has 10 million Lagrangian tracers. VPI is enabled to ensure the occurrence of plastic shearing in the brittle parts of the lithosphere. The model evolution stresses the natural bending of the slab under its own weight triggering the spontaneous retreat of the subducting plate (Fig. 7c). Mantle material upwells in the space opened between the two plates. As in the previous example, shear bands form in the upper portion of the subducting lithosphere favoured by the bending.

## 5.3 Continental Rifting

The structural setting of a stretched lithosphere depends on how the extension distributes in the lithosphere. Extension can be uniform or not uniform hence the overall results can be seen in several geodynamic contexts around the world (Huismans and Beaumont, 2003; Huismans et al., 2005; Huismans and Beaumont, 2014).

An extensional model is really easy to set up since the lithological model consists of straight layers while the initial temper-
ature can be described by a conductive geotherm (Fig. 8a). The model consists of a 125 km thick lithosphere, 35 km of crust and a small seed of weak mantle in the lid that helps to localize the deformation. A wet quarzite rheology is used for the crust and a dry-mantle rheology for the mantle. The initial temperature consists of three conductive layers whose temperatures are set at the top and bottom of each one: crust, lid and asthenosphere. Temperatures of 550 °C, 1330 °C and 1520 °C have been assumed for the moho, the base of the lithosphere and the bottom of the model. The Stokes boundary conditions are free slip
for the left, right and bottom edges while the top edge of the model uses a free surface approximation achieved by including a 10 km weak, low viscosity layer. Numerical instabilities of the Stokes solution in the top layer is reduced by enabling the Stikcy-air approximation (Duretz et al., 2011) in the configuration file. An overall horizontal stretching of 3 cmy$^{-1}$ has been assumed and set in the configuration as prescribed velocity for the free slip right and left edges that can also move. The heat equation is solved setting the Neumann condition at the left and right boundaries and the Dirichlet condition, with fixed tem-
peratures of 0 °C at the top and 1520 °C at the bottom edge. The model domain extends for 1200 km along the $x$ axis and 600 km in depth ($yaxis$) and has 10 million of Lagrangian tracers. VPI is enabled to ensure the occurrence of plastic shearing in the brittle parts of the lithosphere. The configuration file for this model, *rift_sym.cfg* is located in the *cfg* folder of the *LARGE* source code repository.

The model evolves with the formation of paired symmetrical shear zones in the lithosphere that accomodates the thinning and
stretching (Fig. 8b). Lid thinning is extreme and at 4.5 My the mantle lithosphere is completely removed under the stretched crust. A non-uniform stretching model can be achieved by setting a low stretching velocity and enabling strain softening (Huismans and Beaumont, 2003). *LARGE* allows to set two values for the cohesion and friction angle parameters used to evaluate the plastic rheologies of rocks. The asymmetric model uses the same configuration of the symmetric one with some minor changes: double values for friction and accumulated strain and an extensional velocity of 0.6 cmy$^{-1}$. The configuration
file for this model, *rift_asym.cfg* is located in the *cfg* folder of the *LARGE* source code repository. Further localization of the extension at different depths can be achieved by separately scaling the viscosity (creep viscosity) of some layers (upper and lower crust, or mantle lithosphere). The model (Fig. 8c) develops an asymmetric mode of stretching where the deformation localizes on the right shear zone that drives the overall crustal stretching. The lid and the crust show two different thinning



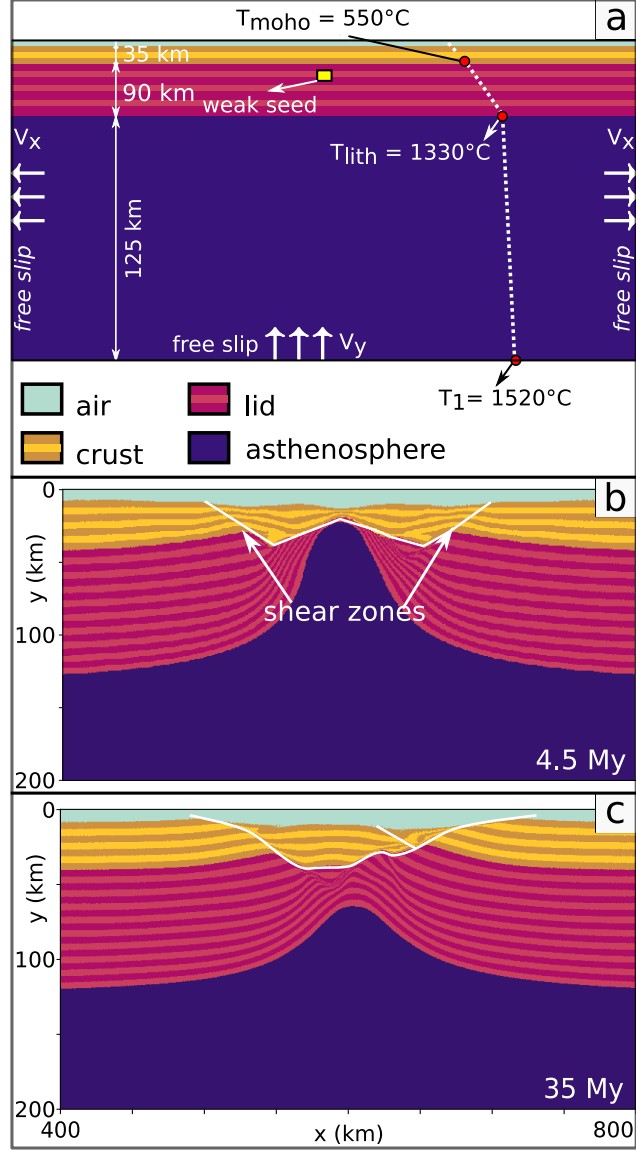

**Figure 8.** (a) Continental extension model setup. Temperature has been fixed to 0 °C at the top, 550 °C at the moho, 1330 °C at the LAB and 1520 °C at the bottom of the domain. An extension velocity of 3 $\mathrm{cmy}^{-1}$ has been split at the left and right edges. (b) Symmetric mode of extension develops after 4.5 My with the formation of two specular shear zones at the left and right of the maximum thinned lid. (c) Evolution after 35 My of the same model with strain softening activated and an extension velocity reduced to 0.6 $\mathrm{cmy}^{-1}$. The position of maximum stretching in the Lid and in the crust is different from above. Here the crustal extension is driven by shear bands at the right of the model.

factors. The positive feedback between increasing strain and strength reduction, due to the decreased internal angle of friction,
cause the asymmetry.





## 6 Parallel scalability

*LARGE* source code has undergone several improvement phases during its development in order to make it as fast as possible. The code has been profiled to find the bottlenecks and the algorithms have been tuned for performance but *LARGE* parallel features depend mainly on third party packages hence much of the parallel scaling resides on them. We arranged a strong

scaling test using the shear bands model described in section 5.1 with 10 million Lagrangian tracers and a 600x600 node grid. The test was run using the GALILEO supercomputer at CINECA. In this test the heat solver and the VPI were deactivated to reduce the CPU usage for each time step and no HDF5 output file was generated to make it independent of I/O. Taking this into account the scaling tests results are promising (Fig. 9). The measured speedup has been fitted to the Amdahl law (Amdahl, 1967):

$$S = \frac{P}{\alpha P + (1 - \alpha)} \tag{14}$$

where $S$ is the speed up, $P$ is the number of parallel processors and $\alpha$ is the serial fraction of the code. From our tests we calculated for this model an $\alpha$ value of 0.06 therefore only 6 percent of the code is serial and a further speed up could only be achieved optimizing this part while any other increase of the number of processor cannot reduce the computing time but on the contrary increases it as we observed.

## 7 Conclusions

We presented here *LARGE*, a finite difference/tracers modeling Python package for the geodynamic community. It has been designed to be flexible and user friendly and developed with advanced modern numerical libraries and algorithms and a structured input system for setting all simulation parameters. All Python modules which make *LARGE*, the command line utilities and the configuration file reference are thoroughly documented in HTML and included in the package together with a complete

set of examples. We have showed, in section 5, the ability of *LARGE* to reproduce results of published studies and models. In the wish list, for the upcoming developments, there are the extension of the supported boundary conditions for the Stokes equation, the use of a petrological model to compute rock density, thermal expansion and heat capacity of rocks, the migration of fluids (water or magma melt) in the lithosphere and their effects.

*Code availability.* LARGE 0.2.0 code is freely available, with a MIT license, from https://bitbucket.org/ncreati/large. The serial version of

*LARGE* depends on the following Python packages: Numpy, Scipy, Numba, QCObj, h5py, ColorAlphabet. The parallel version also depends on: mpi4py, petsc4py, h5pyp and miniga4py. All configuration files used to reproduce examples models of Sect.5 as weel as plotting utilities are included in the repository and in the frozen archive attached to the paper.





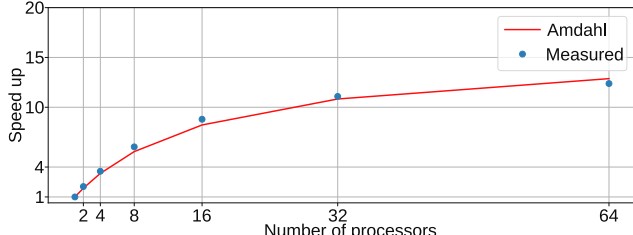

**Figure 9.** Parallel strong scaling for a 10 time steps simulation with different number of processors. The measured speed up has been fitted to the Amdahl law to calculate the fraction of the code that is still serial and that could be optimized to reduce the simulation run time.

*Author contributions.* Nicola Creati designed the application, wrote the paper and prepared the figures. Roberto Vidmar contributed to the writing of the paper, oversaw the drafting of the application documentation, its debug and optimization.

*Competing interests.* The authors declare that the research was conducted in the absence of any commercial or financial relationships that could be construed as a potential conflict of interest

*Acknowledgements.* We acknowledge Adina Pusok for sharing the code used in her paper about velocity interpolation schemes.





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
