# Peer review of "LARGE 0.2.0: 2D numerical modelling of geodynamic problems"

_Geoscientific Model Development, 2020_

## Short Comment (SC1) · 21 Dec 2020

Dear authors,

in my role as Executive editor of GMD, I would like to bring to your attention our Editorial version 1.2:

https://www.geosci-model-dev.net/12/2215/2019/

This highlights some requirements of papers published in GMD, which is also available on the GMD website in the 'Manuscript Types' section: http://www.geoscientific-model-development.net/submission/manuscript_types.html

In particular, please note that for your paper, the following requirement has not been

met in the Discussions paper:

- Code must be published on a persistent public archive with a unique identifier for the exact model version described in the paper or uploaded to the supplement, unless this is impossible for reasons beyond the control of authors. All papers must include a section, at the end of the paper, entitled "Code availability". Here, either instructions for obtaining the code, or the reasons why the code is not available should be clearly stated. It is preferred for the code to be uploaded as a supplement or to be made available at a data repository with an associated DOI (digital object identifier) for the exact model version described in the paper. Alternatively, for established models, there may be an existing means of accessing the code through a particular system. In this case, there must exist a means of permanently accessing the precise model version described in the paper. In some cases, authors may prefer to put models on their own website, or to act as a point of contact for obtaining the code. Given the impermanence of websites and email addresses, this is not encouraged, and authors should consider improving the availability with a more permanent arrangement. Making code available through personal websites or via email contact to the authors is not sufficient. After the paper is accepted the model archive should be updated to include a link to the GMD paper.

Please provide a persistent release for the exact source code version used for the publication in this paper. As explained in https://www.geoscientific-model-development.net/about/manuscript_types.html the preferred reference to this release is through the use of a DOI which then can be cited in the paper.

Yours, Astrid Kerkweg

---

## Short Comment (SC2) · 22 Dec 2020

Dear Editor, we already added our code as supplement archive to the submission as suggested by the Topical Editor (Sylwester Arabas). In the "Code availability" section (line 387) we wrote about it. If this is not enough we can upload it to a data repository with an associated DOI (Zenodo?). Best regards, Nicola Creati

---

## Referee Comment (RC1) · John Mansour (Referee) · 10 Feb 2021

General comments:

Hello Nicola & Roberto. My name is John Mansour and I'm part of the Underworld development team. I'm more a numerical developer than geophysics modeller, and this review will reflect that.

The authors gives an overview of the LARGE Python geodynamics modelling software. LARGE uses a 2D finite difference method with Lagrangian particles to track material evolution and can operate in serial or MPI parallel. It provides methods for incompressible Stokes solutions, and advection-diffusion solution. LARGE provides a rheological model for visco-elasto-plastic behaviours for which the user can set a range

of parameters as necessary. The authors of LARGE should be commended for being quite usability minded, an often overlooked aspect of software design. They achieve this through features such as easy documentation access, a GUI and visualisation tool, logging tools, and a framework for ensuring consistent units usage.

While LARGE appears to be a quality package and will be a welcome addition to the Python geoscience ecosystem, I believe there are some shortcomings in this paper that need to be overcome before it can be recommended for publication. The biggest issue is that the authors haven't given any evidence to support the fidelity of their software. A number of models are presented, but all I can say about these models is that they look qualitatively reasonable. However, we are almost completely in the dark with respect to quantitative accuracy (they do mention shear band angles being reasonable, but that is all). Things that should be done include: * Resolution tests to show behaviour is convergent. * Comparison with analytic tests, and convergence order analysis. * Comparison with results from the literature.

For the analytic tests, Underworld2 provides a suite of tests to ensure that the Stokes system is generating consistent results. SolCx in particular is a standard test most geodynamics software papers will demonstrate. Here are some examples:

https://github.com/underworldcode/underworld2/blob/master/docs/examples/10_Analytic%20Solutions.ipynb
https://github.com/underworldcode/underworld2/blob/master/docs/test/Analytic%20Soln%20Convergence%20Tests.ipynb

Some published results you might consider comparing with: - Blankenbach et al (1989), 2D thermal convection - vanKeken et al (1997), 2D Rayleigh-Taylor and thermochem evolution - Moresi & Solomatov (1998), 2D mantle convection with a brittle lithosphere - Crameri et al (2012), surface topography, sticky air. - Tosi et al (2015), 2D viscoplastic thermal convection - Farrington et al (2014), Viscoelastic shear. - Duretz et al (2011) and Schmid & Podladchikov (2003), circular inclusion - Lemiale et al (2008), shear banding in plastic models - OzBench (2008), slab subduction.

The other main issue with this paper is that the English is too casual. While I understand that English is not the first language of the authors, and I have provided some alternate phrasing suggestions in places, perhaps finding someone to proof read would help expedite this process. There are also simple spelling mistakes that a spell checker should have picked up.

As outlined below, I believe this paper would also be well served to always give the reader enough information to have a basic idea about the different algorithms chosen to be used within LARGE, instead of only providing a reference and assuming that the reader is familiar with that work. The reader shouldn't *need* to access that reference to have basic understanding of what the algorithm entails.

I have outlined a bunch of further smaller issues in the sections below, but one other thing I'll mention is that you should let us know what are the skill requirements necessary to use LARGE. For example, do users need to know Python?

I hope this has been helpful and look forward to seeing the next iteration of this paper. John

Scientific Questions/Issues:

L4-5: "LARGE uses advanced modern numerical libraries and algorithms but unlike common simulation code written in Fortran or C this code is written entirely in Python." I'd be a bit more careful about the wording here. While LARGE is written entirely in Python, most of the libraries LARGE uses will be written in C/Fortran/etc. Also, you should mention some of the other Python based geodynamics codes currently available (UWGeodynamics/Underworld2/PyLith/etc) and how LARGE differs from these.

L18-20: "Geodynamic simulation can be accomplished using commercial software (Pascal and Cloetingh, 2002; Jarosinski et al., 2011), but there are also several other tools available in the literature, both open and closed source, that do the same job and add new numerical methods and algorithms." I think the point to make about open source is that it is good for open science because it provides transparency into the

underlying implementations, and also reproducibility because people can easily re-run the computational experiments (which might not be the case for closed-source and/or commercial software).

L29: Python is perfect suited to large datasets too! Not just small and medium. Check 'dask' for example.

L33-34: "The language is self-sufficient, comes out-of-the-box ready to use, with everything that is needed." What does this mean? I'm not sure it's a useful statement.

L39-50: I'd suggest simplifying this section greatly. I think your target audience should be very well aware of most/all of this. You might simply say "LARGE has been developed on Linux desktop computers, although for larger simulations it may also be deployed to run across multi-node HPC systems."

l50: You make some claims here about Python operation in parallel. Perhaps you can provide some examples to clarify what you are saying here, and compare with other languages for contrast.

l66-67: I think you need to be a bit more specific here. Which finite difference scheme? What order is it? Forwards/backwards/centred/etc? You provide references which is good, but I think your readers shouldn't need to dive into those references to have a clear idea what you're doing. Similarly, what does a staggered mesh mean? What do you mean by regular or irregular? Perhaps an diagram here wouldn't hurt to spell out more succinctly how your discretisation works.

l67-68: Similarly, you'll need to spell out how you are treating the energy equation. Is it a Semi-Largrangian approach, or do your temperature unknowns truly advect freely with the flow?

l73-74: Again, give your reader at least a quick overview of what CorrMinMod is all about, and why it is useful for the class of models you are targeting.

l96-97: "where F is a dimensionless correction term that depends on the type of experiment used to measure the rock rheological parameters and lets to transform the equation in tensorial format". Can you clarify what you are saying here?

l103: Again, specify which iterative scheme you are using to handle the non-linearity. Picard? Newton?

l108: It doesn't follow from unconditional stability that iteration isn't required, although unconditional stability allows for arbitrary time-step sizes.

l110: Again, you need to provide enough detail for your readers to understand what 4-cell or 1-cell is. This isn't standard nomenclature, so if you include it you'll need to explicitly explain what it is. Further, this isn't a bilinear-interpolation scheme. It is a weighted averaging scheme which relies on bilinear-shape functions to determine the weighting.

l132: I'd probably leave details about decorators out, as it's a bit Python language specific, and non-Python readers won't have any idea what it's about.

l135: 'colorAlphabet' seems a bit out of place here as it's not really a scientific concern, *unless* you go on to talk about usability aspects of LARGE and how this is helps to make model development easier.

l154: I'm not sure I'm convinced that this should be listed as distinct from h5py. If it's simply h5py bundled with the necessary libhdf5 libraries, then as far as the paper is concerned it's just h5py.

l189: Give a quick overview of the Halton distribution and why it's used. Diagrams are always nice.

l206: You should give a reference for PETSc. I'd also be interested in the performance differences between the SciPy solver and the PETSc solver (in serial), but perhaps you'll touch on this later.

l219-221: Can you clarify this sentence? Does it really become ill-posed (as in, no

solution exists), or does convergence simply become difficult? What do you mean by "iterating on the domain grid nodes"? l234: I'd probably suggest avoiding speaking about specific parameters (such as "tracersReseedGlobal") as these are more appropriate for your user documentation, and are also quite likely to change as LARGE evolves.

l239: Similarly, perhaps better to omit Figure 3, as these details are almost certain to change in newer versions of LARGE. It's enough to say what you've said l238-240

l244: I'm a bit unsure what you're saying here. Perhaps something like this is sufficient? "LARGEP uses an MPI friendly logging system that minimised user difficulty during the model development stage."

l246-268: Command line utilities: I'd be inclined to think a lot of the details in the section are best left for a LARGE user manual or other similar documentation. I think it's fine to speak about the GUI, and other tools that you have generated to assist the user experience, but discussion should be centred on why it is important for the user experience, rather than the specific tool itself. For example, "To facilitate rapid model development, user documentation for LARGE is made readily available browseable HTML pages proving reference for configuration file specification, Python API, examples, etc etc". Some of the nitty gritty (such as 'large2largep', 'largep') isn't really appropriate here I think, so think to simplify as much as possible.

l270+: Examples. Again, I think you should omit details about configuration files and parameter specifications. This paper should be more focused on the science, not the API. So for example you say: "This simulation does not take in account the temperature therefore the heat calculation is switched off in the configuration file by setting the equation solver to none." where you probably should simply say "This simulation does not take into account temperature effects".

l273: Shear bands. How does your 26 degree shear band angle compare to the results of Kaus (2010)? I believe Moresi et al (2007) also performs similar numerical experiments. Are these results convergent with respect to resolution? What resolution were the simulations performed at? How many particles?

l295: Plates converge: What resolution was used? Particle counts? How does this compare to the results of Gorczyk et al 2007? It looks qualitatively reasonable, but why should we believe this?

l314: Slab retreat: Same questions.. resolution/particles? Comparison with other results? There's no shortage of similar models run using Underworld1/Underword2/other. We want to know whether your results are in line with other published results, and if not then why.

l330: Continental Rifting. Again, it all looks reasonable, but you'll need to convince us that it should be believed.

l364: Which parallel packages? Petsc4py? Anything else? Perhaps specify where most of the parallel time is going.

l370: Fit to this curve will only be true if we assume perfect parallel operation (for the non-serial parts of operation). Of course, this is never truly going to occur due to communicational overheads etc. It might be useful to provide a table or graph showing how execution time is spread across the different simulation phases, and show which phases are serial and which are parallel. Also, i'd be interesting to know how LARGE compares to LARGEP when LARGEP is running using a single processor.. ie PETSc solves vs SciPy solves. Also, even laptops these days have 4+ cores.. how does running on a laptop/desktop utilising multiple cores compare to serial operation?

Technical Corrections:

L5-6: Suggestion: "Simulations are driven by configuration files that define the lithologies and parameters for a given model."

L6-8: Suggestion: "The package can be used to reproduce results of published studies and models or to experiment with new simulations. LARGE can run in serial mode on

desktop computers but can also take advantage of MPI to run in parallel on multi node HPC systems."

L13: Full stop (not comma) required after "measured".

L22-23: Suggestion: "All these tools are written either in Fortran, C or C++ and integrate with well established libraries that provide numerical methods for the solution of differential equations"

L29-31: Suggestion: "The main reasons for Python's success are that it is simple, free, interoperable with other languages, object oriented, and has a large scientific community of users and developers continually extending its potential capabilities."

L31-32: Suggestion: "Since its introduction, several packages have been developed to improve its numerical computation capabilities, in particular Numpy and SciPy (Walt et al., 2011; Virtanen et al., 2020).

L34-35: Suggestion: "Being an interpreted language, it is inherently slow relative to compiled languages, but this limitation can be avoided with careful implementation of the required algorithms."

l100-101: Suggestion: "... the enucleation of shear deformation zones."

l101-102: Suggestion: "The Stokes equations is non-linear and the solution process is not guaranteed to converge (Spiegelman et al., 2016). The non-linearity is owing to the deviatoric stress (eqn 2) which is dependent on a viscosity coefficient which is itself dependent on the velocity and/or pressure solutions."

l108: Cranck -> Crank

l120: Suggestion: "It is the standard implementation for numerical array structures in Python, and its universal adoption greatly enhances interoperability within the scientific Python ecosystem. Leveraging Numpy's vectorized methods in place of standard Python for-loop constructs allows for run-time execution which largely avoids the

Python interpreter. This provides dramatic improvements in numerical throughput, although code readability is at times impacted.

l169-170: Suggestion: "LARGE supports the most commonly used boundary conditions within the geodynamics literature. The Stokes system may be configured for free-slip and no-slip conditions."

l180: "nodes density" -> "node density"

l190-191: Suggestion: "Utilising Numpy and MPI (mpi4py?), this object takes handles all communication between the processors for tracer distribution, advection, deletion, repopulation and ghosting."

l270-272: Suggestion: "The ability of LARGE to solve geodynamics problems can be tested by running standard community models and comparing with published results from within the literature. These models are also provided with LARGE for users to familiarise themselves with the LARGE modelling paradigms."

l303: "Stikcy" -> "Sticky"

l342: "Stikcy" -> "Sticky"

l364: "... but parallel scalability of LARGE is largely dependent on the scalability of the third party dependencies utilised for parallel operation."

l368: Suggestion: ".. promising" -> ".. encouraging".

l386: "weel" -> "well"

---

## Referee Comment (RC2) · David Whipp (Referee) · 10 Feb 2021

**Summary**

Creati and Vidmar present a new 2D geodynamic numerical modelling software package written in Python. The package enables users to run geodynamic experiments with elastic, viscous, and plastic rheologies, offers a flexible way to define model geometries, can be run in serial or parallel, and is open source and freely available. In the manuscript, the authors provide an overview of how the software has been designed, the fundamental equations that are solved, the Python packages it utilities, and some examples of various experiments for different geodynamic scenarios. Overall, the manuscript appears to be a good fit for Geoscientific Model Development, and

the software will be appealing to users given it is free, appears easy to use, and the use of Python may make modification of the software less challenging than with other languages. However, I also feel that there are a handful of issues that should be considered before the manuscript is accepted. Specifically, it would be helpful for the authors to include some demonstrations of the precision/accuracy of the code using some standard benchmark analytical solutions, as well as more clearly presenting what is new/different about this code and how it differs from other existing software. Below I provide a list of the main issues to consider, as well as some detailed remarks that will hopefully assist in revising the text.

**Main comments**

1. The first notable issue in the current version of the text is that there is a limited case made for what is unique about this software. The authors make a point about the code being written in Python, but seem to present more of a case for why Python is an appealing programming language, rather than why it is beneficial to use a code written in Python. I feel that it would be better to emphasize what users gain by using this software, perhaps even including an example of a simple Python function or structure in order to highlight the readability and potential for new users to modify the software. In addition, there is very little comparison to currently available modelling software and how this software differs/improves on those models. It would be helpful for readers to know what other software is being used for this kind of modelling and some of the drawbacks of that software in order to understand why LARGE could be better. It would also be good to emphasize parts of LARGE that are unique and/or better than the earlier software packages, such as ease of visualization, installation, etc. Basically, the authors describe the code, but don't make a very clear case for why users should consider it (other than being a Python code).

2. It would also be helpful to provide some benchmarks to demonstrate that the software reliably reproduces known solutions. These could be in an appendix, or possibly as a main part of the text, but it is important for readers to see that the heat transfer

solution is correct, that plastic shear bands form in the expected locations for plastic problems with analytical solutions, or that the viscous flow pattern agrees with known solutions for channel flow, for example. The authors did a nice job mentioning that there are features in the software to ensure numerical errors are kept under control (time step limitations, etc.), but I don't recall seeing any kind of benchmark to demonstrate the solutions for simple problems are correct and in line with past studies. In particular, it would be good to utilize some 1D heat transfer solutions for different boundary conditions (from Carlslaw and Jaeger, 1959 or Stüwe's Geodynamics of the Lithosphere textbook), viscous flow problems (from Turcotte and Schubert, 2014), or plastic punch problems (see Thieulot et al., 2008, JGR, for example). This would not only allow the authors to demonstrate the accuracy of the code, but also discuss its limitations.

3. If the suggested changes above are made, I would also suggest reducing the number of examples of how the code works that are presented. It is helpful to see a few geological applications of the software, but the description of the results is very brief with the number of examples shown, and the same value for the reader might be gained by showing fewer examples with a bit more detail about the selected cases. Perhaps some examples could be moved to an appendix if the authors would like to still include them.

4. One thing that should be indicated in the text is which versions of the required libraries were used when texting the code for version 0.2.0 of LARGE. It is nice that the authors include the list of dependencies, but some versions may not be compatible with LARGE, and it would help to know which versions have been tested. For example, I had difficulty to install LARGE using Python 3.9 because Numba is currently not compatible with the latest versions of Python 3.

5. Following from the previous point, I was actually unable to get LARGE to run on my Mac running Python 3.7 or 3.8. The installation (following the documentation of the program) was straightforward, but I encountered issues at runtime that prevented me from being able to test the software. As the authors would probably like to know what

happened, I include the log and exception dump from my attempt to run large as an attached zip file.

6. I also note here that I did not find any description of the plotting in LARGE, though it does seem that there are some options for visualization of the output, possibly even using some Python visualization software packages. If not, it could be useful to mention some options for visualizing the data using available tools such as ParaView.

7. Finally, as a style issue there are a number of places where paragraphs seem to be mixing several different points and would benefit from being split into separate paragraphs in order for readers to process the information more easily. I have tried to note some examples below in the specific comments.

**Specific remarks (L=line number)**

Title: Should software be somewhere in the title? For instance "software for modelling geodynamic problems"?

L2: "geodynamic modelling" rather than "geodynamic/modelling"

L3: "large-scale geodynamic...by the finite-difference..."

L4-5: It is not clear what "common simulation code" means here. Can you be more specific? Also, you could already here start making the point about why a Python geodynamic modelling package is useful/helpful.

L6-7: This sentence could be reworded to be more clear.

L7-8: This seems somewhat obvious, is there another point that could be made here about how the code can be used?

L9: Some readers may now know what MPI or HPC are.

L10: As a general comment for the introduction I would suggest trying to clarify problem that you're focusing on. Currently, the first sentence doesn't really say much and could

probably be omitted. Instead, the second sentence makes an important point about the need for models to understand geological systems because of the large range of spatial scales that are considered and the long times over which geological systems evolve. This might be a way to engage readers right from the first sentence.

L21: "he understands" should probably be "they understand"

L22: I am not sure users would be unable to modify software if they do not understand "all the math involved". Perhaps it is better to state only the first point about the programming language.

L26-36: This seems to be more of an advertisement of the nice features of Python rather than what users would gain from using software written in Python. It would be helpful to emphasize the benefits of a Python-based program for users here, rather than why Python has been a success. This could also be a place to compare some of the existing software and the pros/cons of some of the popular modelling codes.

L37-54: This paragraph is fine in general, but it would probably be good to reduce the level of detail in describing LARGE here to make it more accessible to readers. Technical detail is welcome in later sections, but I found it a bit distracting here in the introduction to be presented with code details. Also, this is one paragraph where there are several points mixed together (code details, design limitations for CPUs, etc.)

L57: "scheme" rather than "method"?

L64?: In equation 3 you might want to state somewhere that the velocity term that is normally present for Eulerian solutions is missing because of the Lagrangian reference frame

L65: You refer the readers to Table 1 for the variable names, but I would personally suggest including them here to avoid having readers jump back and forth between the text and table. Most of the variables are using the geoscience conventions, but readers coming from engineering fields may have to refer regularly to Table 1, which is

inconvenient.

L68: Can you list the criterion for solution stability (or cite a reference)?

L70: "increments" rather than "time ranges"

L75: "...Eq. (2) are formulated..."

L97: "...allows transformation of the equation..."

L98: Should you include a reference for the Drucker-Prager criterion?

L100: It could be useful to mention the different rheological behaviors in the Earth earlier (in the introduction?) so that readers understand why there are several different rheological options in LARGE.

L100-103: These sentences should be reworded because it is not clear where the nonlinearity comes from as described.

L111: It seems perhaps there are some features that are not described here, such as strain softening/weakening. If that is an option in LARGE, please describe it in this section.

L118: As mentioned above, it would be good to know which dependency versions were tested for LARGE 0.2.0.

L120: "It is" rather than "It's"

L124: Is there a more quantitative word that could be used here instead of "awesome"?

L130: Some readers may not know what "I/O" means

L149: "...data structure."

L163: "Thermal model" might be better than "Geothermic model"

L164: Here you have a list of heat transfer processes (e.g., conduction) and heat transfer models (e.g., half-space cooling model). You should be careful to distinguish

between them to avoid confusion.

L169: "Currently" rather than "By now"

L175-176: This is a nice feature :)

L187: "global array objects"?

L190: "...regarding the tracers..."

L194: What is the convergence criterion for plasticity in Figure 2? This would be good to present somewhere.

L206: "...uses the Scipy..."

L213: Isn't the particle movement just based on the velocity solution? Might be simpler to say that.

L220: "...overcome by iterating..."

L226: "viscous viscosity" is somewhat confusing. Is there another term that could be used?

L261-264: This is a suggestion for LARGE in general, not specific to the manuscript. Have you considered hosting the documentation on https://readthedocs.org/? It might be nice to have the documentation online and versioned, in case you've not thought about it.

L281: Are your results similar to those in Kaus, 2010?

L286: Is there another word that could be used here instead of "tricky"?

L303: "Sticky-air"

L304: "The left, right...". Also, it might be helpful for readers to distinguish between faces of the model that are fixed in Eulerian space and those with no slip along the face. I found myself a bit confused about what you meant when saying different sides

of the model can move. Is that free slip, of motion of the boundary of the model?

L311: Is the "standard plate model" the half-space cooling model? If so, why not say that instead?

L318: Again, no slip on faces, or no movement of model boundaries?

L334: "horizontal" rather than "straight"

L335: "This model..."

L342: "Sticky-air"

L351: Strain softening is not described in the description of the code.

L361: I would suggest moving this section to be prior to the examples.

L366: CINECA is not familiar to me, and may be unfamiliar to other readers.

L380: Have you demonstrated that LARGE reproduces earlier results? It seems there was no detailed comparison between the results from LARGE and those of earlier studies.

L381-383: This is perhaps something better presented in another section, as it is not really a conclusion. Perhaps if there was a section on future work or limitation of the code you could put this text there.

L386: "well" instead of "feel"

Figure 5: It might be helpful to have consistent color ranges between the panels of Figure 5b in order to see how the deformation localization develops.

Dave Whipp

Please also note the supplement to this comment:
https://gmd.copernicus.org/preprints/gmd-2020-372/gmd-2020-372-RC2-supplement.zip